# Organic dry pea (*Pisum sativum* L.) biofortification for better human health

**Dil Thavarajah**[1]*, **Tristan J. Lawrence**[1], **Sarah E. Powers**[1], **Joshua Kay**[1], **Pushparajah Thavarajah**[1], **Emerson Shipe**[1], **Rebecca McGee**[2], **Shiv Kumar**[3], **Rick Boyles**[4]

**1** Plant and Environmental Sciences, Pulse Quality and Nutritional Breeding, Biosystems Research Complex, Clemson University, Clemson, South Carolina, United States of America, **2** USDA Agriculture Research Service, Grain Legume Genetics and Physiology Research Unit, Washington State University, Pullman, Washington, United States of America, **3** Biodiversity and Crop Improvement Program, International Centre for Agricultural Research in the Dry Areas (ICARDA), Rabat, Morocco, **4** Plant and Environmental Sciences, PeeDee Research and Education Center, Florence, South Carolina, United States of America

* dthavar@clemson.edu

**Data Availability Statement:** All relevant data are within the manuscript and its Supporting Information files.

**Funding:** Funding support for this project was provided by the Organic Agriculture Research and

## Abstract

A primary criticism of organic agriculture is its lower yield and nutritional quality compared to conventional systems. Nutritionally, dry pea (*Pisum sativum* L.) is a rich source of low digestible carbohydrates, protein, and micronutrients. This study aimed to evaluate dry pea cultivars and advanced breeding lines using on-farm field selections to inform the development of biofortified organic cultivars with increased yield and nutritional quality. A total of 44 dry pea entries were grown in two USDA-certified organic on-farm locations in South Carolina (SC), United States of America (USA) for two years. Seed yield and protein for dry pea ranged from 61 to 3833 kg ha$^{-1}$ and 12.6 to 34.2 g/100 g, respectively, with low heritability estimates. Total prebiotic carbohydrate concentration ranged from 14.7 to 26.6 g/100 g. A 100-g serving of organic dry pea provides 73.5 to 133% of the recommended daily allowance (%RDA) of prebiotic carbohydrates. Heritability estimates for individual prebiotic carbohydrates ranged from 0.27 to 0.82. Organic dry peas are rich in minerals [iron (Fe): 1.9–26.2 mg/100 g; zinc (Zn): 1.1–7.5 mg/100 g] and have low to moderate concentrations of phytic acid (PA:18.8–516 mg/100 g). The significant cultivar, location, and year effects were evident for grain yield, thousand seed weight (1000-seed weight), and protein, but results for other nutritional traits varied with genotype, environment, and interactions. "AAC Carver," "Jetset," and "Mystique" were the best-adapted cultivars with high yield, and "CDC Striker," "Fiddle," and "Hampton" had the highest protein concentration. These cultivars are the best performing cultivars that should be incorporated into organic dry pea breeding programs to develop cultivars suitable for organic production. In conclusion, organic dry pea has potential as a winter cash crop in southern climates. Still, it will require selecting diverse genetic material and location sourcing to develop improved cultivars with a higher yield, disease resistance, and nutritional quality.

Extension Initiative (OREI) (award no. 2018-51300-
28431/proposal no. 2018-02799) of the United
States Department of Agriculture, National Institute
of Food and Agriculture (DT and RB), and the
USDA-ARS Pulse Health Initiative (DT); the
Specialty Crop Block Grant Program of the U.S.
Department of Agriculture through grant
AM180100XXXXG026 (DT); the USDA National
Institute of Food and Agriculture, [Hatch] project
[1022664] (DT); and the Good Food Institute (DT).
The funders had no role in study design, data
collection, and analysis, decision to publish, or
preparation of the manuscript. Its contents are
solely the responsibility of the authors and do not
necessarily represent the official views of the
USDA.

**Competing interests:** The authors have declared
that no competing interests exist.

# Introduction

Organic agriculture production has increased since the American Organic Foods Production
Act of 1990. The United States Department of Agriculture (USDA) National Organic Stan-
dards Board describes organic agriculture as "*an ecological production management system
that promotes and enhances biodiversity, biological cycles, and soil biological activity*" [1]. Pulse
crops, including dry pea (*Pisum sativum* L.), increase the ecological, economic, and social ben-
efits of organic cropping systems via biological nitrogen (N) fixation, enhanced biodiversity,
and creation of healthy food systems that can combat malnutrition and obesity. Organic agri-
culture is perceived as more environmentally friendly and sustainable than high-yielding con-
ventional farming systems. Several studies support that notion, indicating organic farming
systems provide a range of soil, biological, ecological, and other environmental benefits over
conventional farming systems [2–4].

Dry pea is an excellent source of complex carbohydrates, protein, vitamins, and minerals
[5, 6]. Dry peas are naturally rich in iron (Fe: 4.6–5.4 mg/100 g), zinc (Zn: 3.9–6.3 mg/100 g),
and magnesium (Mg: 135–143 mg/100 g). In addition, dry pea is naturally low in phytic acid
(PA) (4.9–7.1 mg/g of PA or 1.4–2 mg/g of phytic-P) despite very high total phosphorus (P)
concentrations (3.5–5 mg/g) [5–9]. Nutritionally, dry pea is a rich source of low digestible car-
bohydrates (12–15 g/100 g), protein (20–25 g/100 g), and essential amino acids (e.g., lysine
and tryptophan) [8, 10]. In a symbiotic relationship with Rhizobium bacteria, dry peas fix
atmospheric N, providing 75–120 kg of N per hectare for use by subsequent crops [11].

Consumer demand for pulses has increased due to the demand for plant-based protein
[12]. However, organic farming systems face three significant global challenges: (1) maintain-
ing crop productivity to produce enough food for a projected population of 9 billion in 2050,
(2) delivering the expected nutritional quality as a human food and animal feed, and (3) main-
taining ecological benefits, e.g., N and P use efficiency [13]. A primary criticism of organic
agriculture is lower yield and nutritional quality compared to non-organic systems. Organic
grains use soil nutrients derived from organic cover crop breakdown. Organic consumers
believe organic foods are nutritionally superior and improve human health compared to con-
ventional foods; however, organically grown grains typically have lower yields and nutritional
quality than conventionally grown crops [2, 14, 15]. A meta-analysis of over 10,000 organic
farmers representing >1.9m acres of organic farmland demonstrated that averaged among
food crops [wheat (*Triticum aestivum* L.), maize (*Zea mays* L.), common bean (*Phaseolus vul-
garis* L.), potato (*Solanum tuberosum* L.), and vegetables], the organic yield was 80% of conven-
tional yield [16]. The organic to conventional yield ratio varied with crop type, cultivars in
production, and growing locations, highlighting the importance of regional breeding pro-
grams for organic production [17]. Therefore, it is essential within the organic farming frame-
work to focus on organic plant breeding, resulting in more suitable cultivars for organic
production and delivering enhanced nutritional quality and nutrient bioavailability to combat
micronutrient malnutrition, obesity, and overweight.

Current world pea production is 14.1 MMT on over 18 million acres, with US dry pea pro-
duction representing about 7.1% of world production on 1,052,001 acres [18]. The USDA does
not report definite statistics on organic dry pea acreage. Still, acres devoted to organic pulse
crops are approximately 1.5–2% of dry pea and lentil acreage. In 2011, certified organic dry
peas and lentils were grown on more than 17,877 acres; North Dakota and Washington led
with over 3,500 acres each [19]. Yellow dry pea has become one of the popular cool-season
legumes grown in South Carolina (SC) during the winter. In the Pee Dee region, Carolina soils
have pH and soil P, potassium (K), and organic matter levels appropriate for dry pea germina-
tion, establishment, and growth. A rotational cropping system of dry pea and cereal has shown

promise in sustainable, non-organic farming operations [20]. Winter legumes provide weed control and available soil N and P for the following summer grain crop [8]. Developing crops for optimal performance in organic management systems requires integrating a range of traits, such as yield, agronomy, nutrient use efficiency, disease resistance, and nutritional quality. However, no breeding efforts have aimed to reduce the yield gap or increase dry pea's nutritional quality (i.e., biofortification) for organic farming systems. Similarly, genomic and translational resources for selecting dry pea cultivars for organic production are also nonexistent.

Biofortification is a sustainable approach using conventional plant breeding and molecular techniques to improve dry pea nutritional quality [21–23]. Most of the pulse breeding programs globally use micronutrient biofortification with the key targets of Fe, Zn, carotenoids, and folates. In recent years, HarvestPlus, with global partners in Africa and Asia, released several biofortified pulse cultivars to combat micronutrient malnutrition [22, 24–26]. Most biofortification breeding efforts were made in the conventional farming system. Still, organic biofortification research is yet to be conducted. With increasing societal nutritional needs for organically grown dry pea, biofortification brings organic plant breeding and nutritional sciences together to work on the persistent problems of human nutrition. In addition, biofortification of dry pea under organic systems will improve human nutrition, provide N and carbon (C) benefits to subsequent cereal and vegetable crops, and increase nutrient use efficiency and biodiversity [27, 28]. Current organic pulse production depends on cultivars that have been bred for non-organic production, but these are often not suited to organic production. For example, these cultivars may have a low grain yield, production issues (weed control, disease resistance, etc.), and low nutritional quality. The testing hypothesis of this study was to evaluate if the current dry pea cultivars and advanced breeding lines in production vary in grain yield and nutritional quality with response to the organic cropping systems. The objectives of this study were to evaluate 44 dry pea entries in two on-farm locations for two years to determine grain yield and nutritional quality for human food, e.g., high protein, low digestible carbohydrates, and minerals as well as low phytate.

## Materials and methods

### Materials

Standards, chemicals, and high-purity solvents used for prebiotic carbohydrate, minerals, and PA analysis were purchased from Sigma Aldrich Co. (St. Louis, MO), Fisher Scientific (Waltham, MA), VWR International (Radnor, PA), and Tokyo Chemical Industry (Portland, OR) and used without further purification. Water, distilled, and deionized (ddH$_2$O) to the resistance of ≥18.2 MΩ×cm (PURELAB flex 2 system, ELGA LabWater North America, Woodridge, IL) was used for sample and reagent preparation.

### Experimental details

The experimental field design was a randomized complete block design (RCDB) with 44 dry pea entries (25 cultivars and 19 advanced breeding lines) with two replications at two locations in 2019 and three replications at one location in 2020 (n = 308; **Table 1**). Due to the heavy rain and flooding, the Clemson field location was not planted in 2020. The commercial dry pea cultivars were purchased from Pulse USA (Bismark, ND, USA), Meridian Seeds (Mapleton, ND, USA), and the Washington State Crop Improvement Association (Pullman, WA, USA). The advanced dry pea breeding accessions were obtained from the USDA-ARS Pulse Breeding Program, Washington State University, WA, USA (**Table 1**). Material transfer agreements (MTAs) were signed with the seed companies and the USDA-ARS to test these entries in SC, USA. These dry pea cultivars were selected based on yield potential, disease resistance, and consumer acceptability. Before sowing, two soil samples were randomly taken at 0–15 cm depth from each plot. The soil samples

**Table 1. Experimental design used in the dry pea nutritional breeding trials.**

| Year (location) | 2019 (Clemson; Pelion), 2020 (Pelion) |
|---|---|
| Location | Clemson, SC; Pelion SC |
| Replicates (Year) | 2 (2019); 3(2020) |
| Cultivars/ Breeding lines | **Cultivars (25)**: AAC Carver, AAC Comfort, AC Agassiz, AC Earlystar, Banjo, CDC Amarillo, CDC Gwater, CDC Inca, CDC Saffron, CDC Spectrum, CDC Striker, Delta, DS Admiral, Durwood, Fiddle Flute, Hampton, Jetset, Korando, LG Koda, Matrix, Mystique, Nette 2010, SW Arcadia, SW Midas<br>**Breeding lines (19)**: PS01100925, PS03101445, PS05100735, PS08100582, PS08101004, PS08101022, PS12100047, PS14100079, PS1410B0003, PS1410B0006, PS1410B0065, PS1410B0073, PS1514B0002, PS16100003, PS16100038, PS16100085, PS16100086, PS16100096, PS16100127 |
| Total | 308 |

were homogenized, and three composite samples were analyzed for soil properties at the Clemson University Soil Testing laboratory, SC, USA. For the crop history, the 2019 Pelion location had grass cover crops, and the Clemson location had cereal rye cover crops.

## Land preparation

USDA-certified organic on-farm locations were WP Rawl and Sons (Pelion, SC, USA) and Calhoun Fields Laboratory (Clemson University, SC, USA). Before dry pea planting, the Pelion location had ryegrass cover crop followed by sorghum (*Sorghum bicolor* L.) and kale (*Brassica oleracea* L.), and Clemson location had mixed (legume-cereal-grass) cover crops followed by red beets (*Beta vulgaris* L.) in 2019. In 2020, the Pelion location had kale followed by English peas. Before planting, fields were tilled using a disc harrow and smoothly leveled. All plots were then marked with a weatherproof barcoded field tag, and cultivar "Hampton" was planted as a control to eliminate the border effect. A cone plot planter was used for sowing seed in 1.4×6 m plots (8.4 m$^2$) containing seven rows spaced 20 cm apart, with a seeding depth of 5–7 cm, at a seeding rate of 90 seeds m$^{-2}$. USDA-certified organic inoculant (Peaceful Valley Farm Supply, Inc, USA) was added to the seed packets at the rate of 3.1 g kg$^{-1}$ of seed. Organically certified fertilizers, pesticides, and chemicals were not used in this experiment; weeds were removed by a mechanical cultivator attached to a small tractor. Irrigation was not provided. The Pelion field location was planted on January 29, 2019, and harvested on May 22, 2019; then, the Clemson fields were planted on February 4, 2019, and harvested on May 30, 2019. For the second year, Pelion fields were planted on January 29, 2020, and harvested on June 3, 2020. At physiological maturity (110–115 days after planting), the plots were harvested using a small plot. Subsamples (500–750 g) of harvested seeds were stored at −10°C until nutritional quality analysis. Additional dry pea samples collected from each replication were hand cleaned, finely ground using a UDY grinder, and then stored at −10°C until nutritional quality analysis. All nutritional quality data are reported on a dry basis (15% moisture).

## Thousand seed weight (1000-seed weight)

Dry pea grain yield was calculated based on the size of the plot, and 1000-seed weight was calculated from the weight of 100 seeds, measured using a top-loading electronic balance.

## Protein analysis

Finely ground dry pea samples were sent to the Soil Testing Laboratory, Clemson University, SC, for total N analysis, and then values were converted to total protein content by multiplying by 6.25.

## Prebiotic carbohydrate analysis

Dry pea seeds were ground (Blade Coffee Grinder, KitchenAid, St. Joseph, MI, USA) and sieved to 0.5-mm particle size. Carbohydrates were extracted by the method [29]. Ground dry pea samples (150 mg) were weighed into a centrifugal polypropylene tube (VWR International, Radnor, PA, USA). After adding 10 mL of water, each tube was mixed on a vortex mixer and placed in a water bath for 1 h at 80˚C. Tubes were then centrifuged at 3000 g for 10 min, and the supernatant was filtered through a 13 mm × 0.45 μm nylon syringe filter (Thermo Fisher Scientific, MA, USA) into an HPLC vial. Carbohydrate analysis was done using a Dionex ICS-5000+ system (Thermo Scientific, Waltham, MA, USA) equipped with a pulsed amperometric detector (PAD) with a working gold electrode and a silver-silver chloride reference electrode. Analyte separation was achieved using a Dionex CarboPac PA1 analytical column (250 × 4 mm) in series with a Dionex CarboPac PA1 guard column (50 × 4 mm). Pure standards were used to identify peaks, generate calibration curves, and monitor detector sensitivity; a lab reference sample was also used to monitor extraction consistency. Concentrations were quantified within a linear range of 0.1–500 ppm with a minimum detection limit of 0.1 ppm. Each carbohydrate's concentration was calculated according to $X = (C \times V) / m$, where X is the moisture-corrected analyte concentration in the sample, C is the concentration in the filtrate, and V is the sample volume, and m is the mass of the sample.

## Starch analysis

Resistant starch (RS), non-resistant starch (NRS), and total starch (TS) were measured using the modified Megazyme resistant starch assay method [30]. Samples (100 mg) of finely ground seed were weighed into centrifugal polypropylene tubes, to which an enzyme solution (2 mL) containing amyloglucosidase (3 U/mL) and α·-amylase (10 mg/mL) in sodium maleate buffer (100 mM, pH 6.0) was added. Tubes were then incubated with constant circular shaking (200 strokes/min) for 16 h at 37˚C. Ethanol (4 mL; 99%) was added, then the tubes were vortexed, centrifuged at 1500 g for 10 min, and decanted into 100-mL volumetric flasks. Two additional washings were performed by adding 2 mL of ethanol (50%) and vortex mixing to suspend the pellet, followed by an additional 6 mL of ethanol (50%), vortex mixing, centrifugation, and decanting. Pooled non-resistant starch washings were brought to 100 mL volume with water. Pellets containing resistant starch were dissolved in 2 mL of 2 M potassium hydroxide (KOH) with a magnetic stir bar for 20 min in an ice water bath. Sodium acetate buffer (8 mL, 1.2 M, pH 3.8) was added, immediately followed by 0.1 mL of amyloglucosidase (AMG; 3300 U/mL). Samples were incubated at 50˚C in a water bath for 30 min. Tubes were then centrifuged (1500 g for 10 min). RS and NRS fractions were quantified via spectrophotometry. Starch solution (0.1 mL) and glucose oxidase/peroxidase (GOPOD) reagent (3 mL) were added to glass tubes and incubated for 20 min at 50˚C. A glucose standard (1 mg/mL in 0.2% benzoic acid) was included in each batch. Absorbance was measured at 510 nm against a reagent blank. NRS was calculated using the formula NRS (g/100 g sample) = $\Delta E \times F/W \times 90$, where $\Delta E$ is the absorbance of the sample, F is the absorbance to microgram conversion factor (100 / absorbance of glucose standard), W is the sample dry weight, and 90 includes adjustments for volume, unit conversions, and free to anhydrous glucose. A similar formula was used to calculate RS, RS (g/100 g sample) = $\Delta E \times F/W \times 9.27$, where 9.27 includes adjustments for volume, unit conversions, and free to anhydrous glucose. TS was calculated as TS = RS + NRS.

## Mineral analysis

Dry pea seed minerals were measured using a modified acid digestion method followed by inductively coupled plasma emission spectrometry [31]. Finely powdered 200g of dry pea seed

were digested overnight in 4 mL of concentrated nitric acid (70% $HNO_3$). The seed samples were then heated to 150°C for 2 h, with 4 mL of hydrochloric acid (70% HCl), then added to the solution and heated for an additional 1 h. The digested solution was filtered through Whatman paper (20–25 μm) and diluted to 10 mL with deionized $H_2O$. Mineral concentrations of potassium (K), calcium (Ca), magnesium (Mg), P, Fe, Zn, manganese (Mn), copper (Cu), and selenium (Se) were determined by inductively coupled plasma emission spectrometry (ICP-OES; ICP-6500 Duo, Thermo Fisher Scientific, Pittsburg, PA, USA). Standards made from a 1000 mg $L^{-1}$ stock solution were serially diluted to produce calibration curves from 0.5 to 5.0 mg $L^{-1}$. Ground organic dry pea and National Institute of Standards Technology (NIST) reference peach leaf (SRM 1547) samples were used as a laboratory and standard references for data quality control.

## Phytic acid (PA) analysis

Seed samples were prepared using the modified PA extraction method [32]. A 100-mg sample of finely ground dry pea seed was weighed into a 15-mL conical tube with a fitted cap. Then 10 mL of 0.5 M HCl were added to the tube, which was submerged into boiling water (~100°C) for 5 min. The solution was centrifuged for 3 min; then, the supernatant was transferred with the addition of 1.5 mL of 12 M HCl. High-performance liquid chromatography with a conductivity detector was used for IP6 analysis (ICS-5000 Dionex, Sunnyvale, CA, USA). The PA was separated with an Omnipac Pax-100 guard column (8 μm) and quantified by conductivity detection. The solvents used for gradient elution were 130 mM sodium hydroxide (A), deionized water-isopropanol (50:50, v/v) (B), and water (C). The flow rate of the gradient elution was 1.0 mL $min^{-1}$ with a total run time of 10 min. Retention time and peak area were used to quantify the PA in the seed samples. PA standards from 10 to 500 mg $L^{-1}$ were used for calibration curves, with the detection limit set at 5 mg $L^{-1}$. The error tolerance was <0.1% for all laboratory samples.

## Statistical analysis

Replicates, years, and genotypes were included as class variables. Data from both years were combined (after testing for heterogeneity) and analyzed using a general linear model procedure (PROC GLM) mixed model. Fisher's least significant difference (LSD) at $\leq 0.05$ was performed for mean separation [33]. Correlations (Pearson correlation coefficients) among yield, TSW, and other traits were determined. ANOVA was used to determine if the effect was significant. A statistical model was developed to estimate broad-sense heritability ($H^2$) with the variables and genotype as random effects. The model was calculated using the restricted maximum likelihood (REML) method. $H^2$ was estimated as the proportion of variance due to genotype, and analyses were performed using JMP 14.0.0 and SAS 9.4.

## Results

### Field weather and soil conditions

The field trials took place at Clemson and Pelion, SC, during 2019 and at Pelion, SC, in 2020. A total of 25 cultivars and 19 breeding lines were evaluated at each location, with two replicates in 2019 due to seed limitations and three replicates in 2020 (n = 308) (**Table 1**). In 2019, the Pelion, SC location was warmer (25.6°C) and received more precipitation (68.6 mm) in May than the Clemson, SC location. In 2020, the average temperature was lower (20.8°C), and the average precipitation was higher (236 mm) at Pelion, SC, than in the previous year (**Table 2**). In 2019, the Clemson field had a lower pH (6.3), with higher N-nitrate [(N-$NO_3$): 48 ppm], K

**Table 2. Mean monthly temperature and precipitation for two growing locations in SC, USA.**

| Year | Location | Source | Jan | Feb | Mar | Apr | May |
|------|----------|--------|-----|-----|-----|-----|-----|
| 2019 | Clemson | Temp (˚C) | 6.1 | 10.0 | 10.8 | 16.9 | 23.1 |
| | | Precipitation (mm) | 140 | 193 | 88.9 | 117 | 19.3 |
| | Pelion | Temp (˚C) | 9.4 | 12.8 | 13.6 | 19.4 | 25.6 |
| | | Precipitation (in) | 3.6 | 1.7 | 2.6 | 4.3 | 2.7 |
| 2020 | Pelion | Temp (˚C) | 9.6 | 11.0 | 16.6 | 17.6 | 20.8 |
| | | Precipitation (in) | 69 | 172 | 83 | 81 | 236 |

(284 lbs/ac), and organic matter (4.3%) than the Pelion field, which had more P (727 lbs/ac). In 2020, Pelion soil values reflected higher pH (6.8 to 7.1), N-NO$_3$ (16 to 21 ppm), and organic matter (0.8 to 1.1%) compared to 2019, as well as lower levels of P (727 to 549 lbs/ac) and K (108 to 81 lbs/ac) (**Table 3**). Clemson soils are clay loam, and Pelion soils are sandy, explaining the differences in N, K, and organic matter.

## Analysis of variance

Cultivar was significant at *P<0.05* and *P<0.1* for all traits except for maltose, RS, TS, and Se (**Table 4**). For yield, cultivar, year, and cultivar × location were highly significant at P<0.05, location and cultivar × year were significant at *P<0.1*, and all components were highly significant (*P<0.05*) for the 1000-seed weight (**Table 4**). Only cultivar × location was not significant for protein, with all other components highly significant (*P<0.05*) (**Table 4**). Broad-sense heritability estimates indicated 1000-seed weight was more heritable (H$^2$ = 0.69) than yield (H$^2$ = 0.21) and protein (H$^2$ = 0.24). Most prebiotic carbohydrates varied with dry pea cultivar except for maltose and starch polysaccharides. For sugar alcohols, the location was not significant for xylitol and mannitol, the year was not significant for sorbitol, cultivar × location was not significant for mannitol, and cultivar × year was not significant for sorbitol; all other components were significant (*P<0.05*) for each sugar alcohol (**Table 4**). For simple sugars, only cultivar and location significantly (*P<0.05*) affected glucose concentration, and only location and year were significant (*P<0.05*) for maltose concentration. Cultivar × location was not significant for fructose concentration, and cultivar × year was not significant for sucrose concentration. Location was not significant for arabinose concentration, with all other components being highly significant (*P<0.05*) for simple sugars. For raffinose oligosaccharides (RFO) and fructooligosaccharides (FOS), the location was not significant. For verbascose +kestose (Ver+Kes), and cultivar × location was not significant for nystose, with all other components significant (*P<0.1* and *P<0.05*) for each RFO and FOS (**Table 4**). Location (*P<0.05*), year (*P<0.1*), and cultivar × year (*P<0.05*) had significant effects on RS, while only location and year were significant (P<0.05) for TS. Prebiotic carbohydrates exhibited broad heritability ranges for organic dry pea, with glucose and fructose having the lowest heritability at 0.29 and 0.27, respectively. Galactinol (H$^2$ = 0.74) and Ver+Kes (H$^2$ = 0.75) had the highest heritability, with all other prebiotic carbohydrates having moderate to high heritability, except for maltose and the starch

**Table 3. Soil chemical properties at the locations where dry pea was grown in 2019 and 2020.**

| Year | Location (Soil type) | Soil pH | N-NO$_3$ (PPM) | P (lbs/ac) | K (lbs/ac) | Organic Matter (%) |
|------|---------------------|---------|----------------|------------|------------|--------------------|
| 2019 | Clemson (Clay loam) | 6.3 | 48 | 76 | 284 | 4.3 |
| | Pelion (Sandy) | 6.8 | 16 | 727 | 108 | 0.8 |
| 2020 | Pelion (Sandy) | 7.1 | 21 | 549 | 81 | 1.1 |

**Table 4. Analysis of variance and broad-sense heritability estimates of yield and nutritional traits evaluated for dry pea genotypes tested in SC, USA.**

| Component | Cultivar | Location | Year | Cultivar × Location | Cultivar × Year | $H^2$ |
|---|---|---|---|---|---|---|
| **Yield** | ** | * | ** | ** | * | 0.21 |
| **TSW** | ** | ** | ** | ** | ** | 0.69 |
| **Protein** | ** | ** | ** | NS | ** | 0.24 |
| *Prebiotic carbohydrates* | | | | | | |
| *Sugar Alcohols* | | | | | | |
| Myo-Inositol | ** | ** | ** | ** | ** | 0.52 |
| Xylitol | ** | NS | ** | ** | ** | 0.66 |
| Galactinol | ** | ** | ** | ** | ** | 0.74 |
| Sorbitol | ** | ** | NS | ** | NS | 0.42 |
| Mannitol | ** | NS | ** | NS | ** | 0.57 |
| *Simple Sugars* | | | | | | |
| Glucose | ** | ** | NS | NS | NS | 0.29 |
| Fructose | ** | ** | ** | NS | ** | 0.27 |
| Sucrose | ** | ** | ** | ** | NS | 0.52 |
| Arabinose | ** | NS | ** | ** | ** | 0.65 |
| Maltose | NS | ** | ** | NS | NS | 0.00 |
| *RFO and FOS* | | | | | | |
| Sta+Raf | ** | ** | ** | * | ** | 0.64 |
| Ver+Kes | ** | NS | ** | ** | ** | 0.75 |
| Nystose | ** | ** | ** | NS | * | 0.27 |
| *Starch Polysaccharides* | | | | | | |
| Resistant starch | NS | ** | * | NS | ** | 0.00 |
| Total starch | NS | ** | ** | NS | NS | 0.00 |
| **Minerals** | | | | | | |
| K | ** | ** | ** | * | NS | 0.07 |
| Ca | * | ** | ** | NS | NS | 0.03 |
| Mg | * | ** | NS | NS | NS | 0.00 |
| P | ** | NS | NS | NS | NS | 0.02 |
| Fe | ** | ** | ** | ** | NS | 0.00 |
| Zn | ** | ** | ** | NS | NS | 0.03 |
| Mn | * | NS | NS | NS | NS | 0.00 |
| Cu | ** | NS | NS | NS | NS | 0.00 |
| Se | NS | ** | ** | NS | NS | 0.00 |
| Phytic acid | * | NS | ** | NS | NS | 0.00 |

Raffinose family of oligosaccharides (RFO); Fructooligosaccharides (FOS); Stachyose, and Raffinose (Sta+Raf)

Verbascose and Kestose (Ver+Kes)

** significant at *P<0.05*

* significant at *P<0.1*; Not significant (NS); $H^2$ broad-sense heritability estimate.

polysaccharides, which were not heritable. For mineral concentrations, the cultivar was significant for all minerals except Se; cultivar × location was only significant for K (*P<0.1*) and Fe (*P<0.05*), and cultivar × year was not significant for any mineral (**Table 4**). Location was significant (*P<0.05*) for K, calcium (Ca), magnesium (Mg), Fe, Zn, and selenium (Se) but not for P, manganese (Mn), and copper (Cu). Additionally, the year was significant (*P<0.05*) for K, Ca, Fe, Zn, and Se but not for Mg, P, Mn, and Cu. Finally, only cultivar (*P<0.1*) and year (*P<0.05*) were significant for the PA concentration of organically grown dry pea (**Table 4**). All minerals were found to be not heritable.

## Nutritional quality

Organic dry pea shows broad phenotypic variation for protein (12.6–34.2 g/100 g), prebiotic carbohydrates (12.5–19.8 g/100 g), minerals, and PA (88.8–354 mg/100 g) (Table 5). Organic dry pea can provide a significant portion of the recommended daily allowance (RDA) of prebiotic carbohydrates (81%), protein (38–46%), and a range of minerals (Table 5). Organic dry pea provides a significant amount of the %RDA for K (29.6–38.8%), Mg (31.3–40.3%), Zn

**Table 5. Range and mean nutrient concentrations of organic dry pea grown in SC.**

| Nutrient | Organic | | %RDA | |
|---|---|---|---|---|
| | Range | Mean | Female | Male |
| **Protein (g/100 g)** | 12.6–34.2 | 21.1 | 27-74(46) | 23-61(38) |
| *Prebiotic carbohydrates* | | | | |
| *Sugar Alcohols (mg/100 g)* | | | | |
| Myo-Inositol | 98–399 | 244 | | |
| Xylitol | 2.5–31.7 | 15.7 | | |
| Galactinol | 91.3–425 | 163 | | |
| Sorbitol | 8.4–115 | 34.9 | | |
| Mannitol | 0.9–23.8 | 5.9 | | |
| *Simple Sugars (mg/100 g)* | | | | |
| Glucose | 14.6–137 | 62 | | |
| Fructose | 1.7–30.7 | 6.4 | | |
| Sucrose | 1530–3043 | 2156 | | |
| Arabinose | 3.3–13.1 | 7.2 | | |
| Maltose | 2.1–289 | 26.3 | | |
| *RFO and FOS (mg/100 g)* | | | | |
| Sta+Raf | 2111–4077 | 3128 | | |
| Ver+Kes | 1548–3929 | 2688 | | |
| Nystose | 1.6–9.1 | 3.4 | | |
| *Starch Polysaccharides (g/100 g)* | | | | |
| Resistant starch | 4.2–10 | 7.6 | | |
| Total starch | 35.4–66.9 | 52.6 | | |
| Total known prebiotic carbohydrates *(g/100 g)* | 12.5–19.8 | 16.1 | 63–99 (81) | 63–99 (81) |
| *Minerals (mg/100 g)* | | | | |
| Potassium (K) | 322–1716 | 1008 | 38.8 | 29.6 |
| Calcium (Ca) | 11–338 | 94 | 7.8–9.4 | 9.4 |
| Magnesium (Mg) | 46–232 | 125 | 39.1–40.3 | 31.3 |
| Phosphorus (P) | 123–759 | 377 | 53.9 | 53.9 |
| Iron (Fe) | 1.9–26.2 | 5.7 | 31.7–71.3 | 71.3 |
| Zinc (Zn) | 1.1–7.5 | 3.2 | 40.0 | 29.1 |
| Manganese (Mn) | 0.4–3.4 | 1.2 | 66.7 | 52.2 |
| Copper (Cu) | 0.2–3.5 | 0.8 | 88.9 | 88.9 |
| Selenium (Se: μg/100 g) | 0–130 | 20 | 36.4 | 36.4 |
| Phytic acid (mg/100 g) | 88.8–354 | 159 | | |

Values are based on the combined statistical analysis of 308 data points for the current study (dry weight basis). Total prebiotic carbohydrates include sugar alcohols, simple sugars, raffinose-family oligosaccharides, and resistant starch. % RDA is based on 20 g/day for total prebiotic carbohydrates [30]. %RDA for protein is 46 g/day for women aged 19–70+ years and 56 g/day for men aged 19–70+years. Mineral %RDA values are from the National Institute of Health (https://www.ncbi.nlm.nih.gov/books/NBK545442/table/appJ_tab3/?report=objectonly)

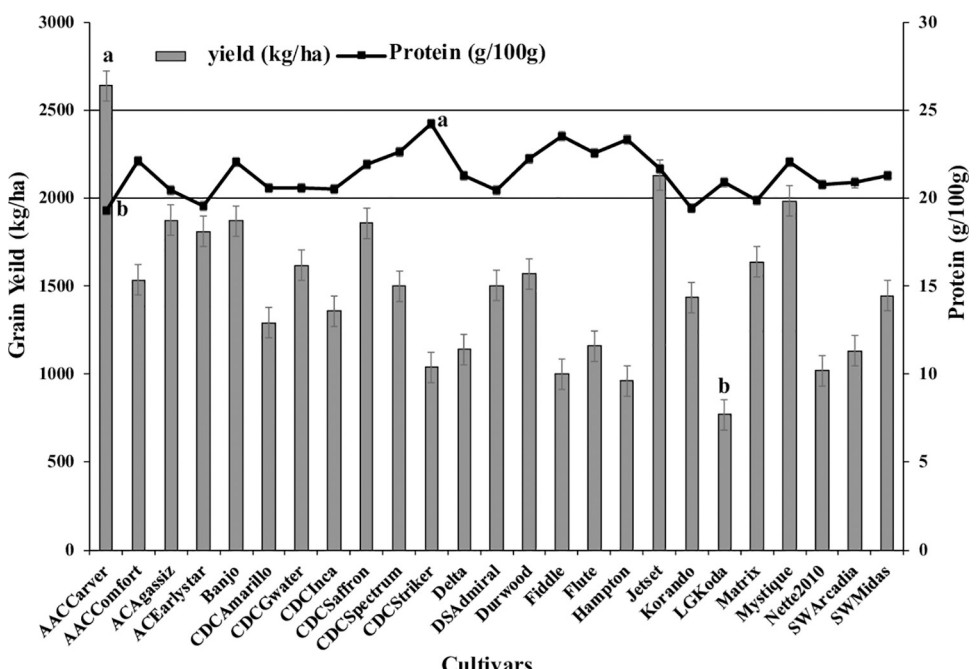

**Fig 1. Variation of grain yield and protein content among dry pea cultivars grown in the organic system.**

(29.1–40%), and Se (36.4%) for both men and women but is not a good source of Ca (7.8–9.4%) in the diet (**Table 5**).

## Cultivar responses

Yield varied among the organically grown cultivars, with "AAC Carver" having the highest yield (~2600 kg/ha) and "LG Koda" the lowest (~750 kg/ha) (**Fig 1**). "AAC Carver" had one of the lowest protein concentrations (~19 g/100g), while "CDC Striker," which had one of the lowest yields (~1000 kg/ha), had the highest protein concentration (~24 g/100 g) (**Fig 1**). Cultivars varied in the total concentrations of the sugar alcohols, myo-inositol, xylitol, galactitol, sorbitol, and mannitol (**Fig 2A**). The cultivar "Hampton" had the lowest concentration of sugar alcohols (~425 mg/100 g) and "CDC Greenwater" the highest (575 mg/100 g) (**Fig 2A**). All cultivars had varying concentrations of RFOs (Raf+Sta and Ver+Kes), with cultivar "Fiddle" having the lowest total RFO concentration (~5200 mg/100 g) and cultivar "Mystique" the highest (~6000 mg/100 g) (**Fig 2B**). Pearson's correlation analysis was performed to determine significant correlations between agronomic and nutritional quality traits (**Fig 3**). A significant ($P<0.05$) and strong correlation was observed for total water-soluble carbohydrates and yield (r = 0.42), with low but significant ($P<0.05$) positive correlations found between 1000-seed weight and yield (r = 0.2), and 1000-seed weight and total water-soluble carbohydrates (r = 0.26) (**Fig 3**). Protein was significantly ($P<0.05$) negatively correlated with all agronomic traits: yield (r = −0.2), TSW (r = −0.26), and total water-soluble carbohydrates (r = −0.1) (**Fig 3**). More specifically, significant ($P<0.05$) negative correlations were found between yield and xylitol, mannitol, sucrose, arabinose, maltose, and RS, but the yield was significantly ($P<0.05$) positively correlated with galactinol, sorbitol, glucose, fructose ($P<0.1$), all RFO and FOS, as well as soluble starch and TS (**Table 6**). Finally, the yield was not correlated with Zn, P, or PA but was positively correlated with both Mg ($P<0.05$) and Fe ($P<0.1$). A significant ($P<0.1$) negative correlation was observed between yield and K (**Table 7**). Positive, significant

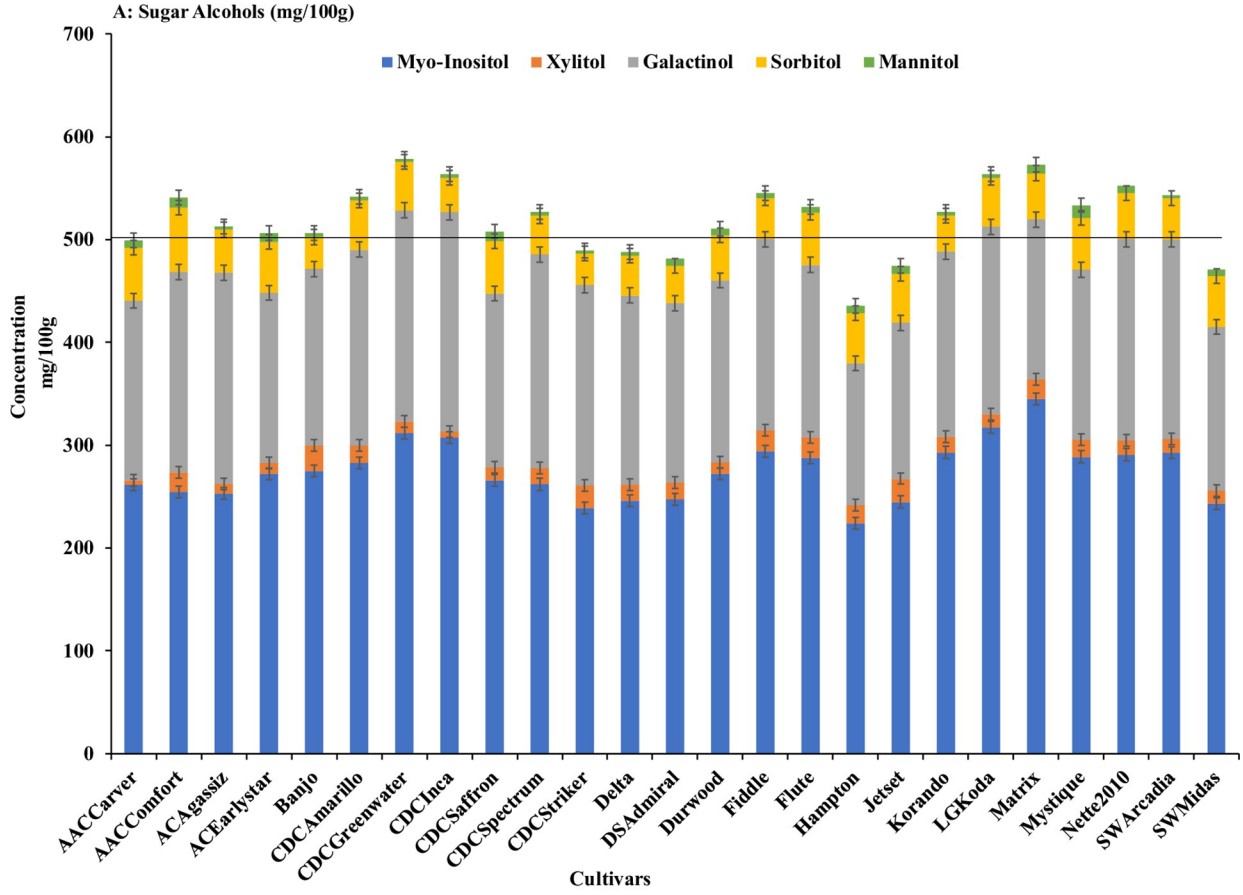

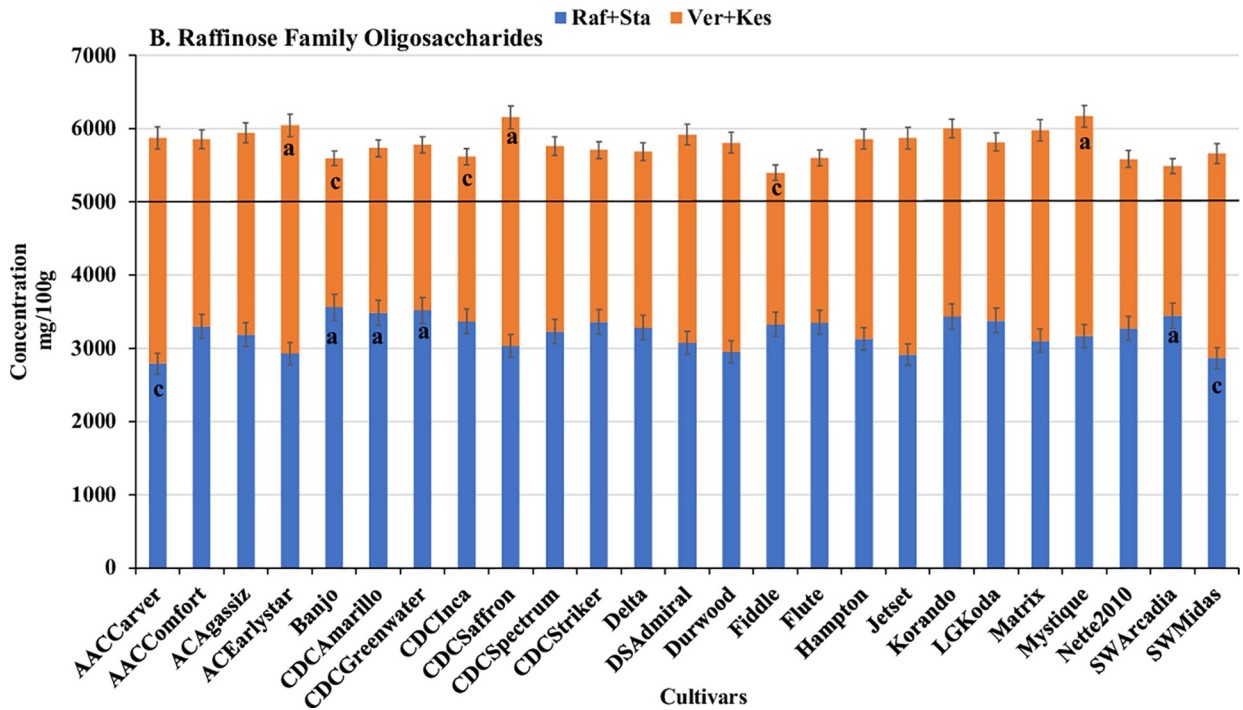

**Fig 2.** Variation of (A) sugar alcohols and (B) raffinose family oligosaccharides concentrations among dry pea cultivars grown in an organic system.

correlations were evident for protein and myo-inositol ($P<0.1$), xylitol ($P<0.1$), mannitol ($P<0.05$), sucrose ($P<0.1$), arabinose ($P<0.05$), and maltose ($P<0.05$). Protein was predominantly negatively correlated with RFO and FOS carbohydrates ($P<0.05$) (**Table 6**). All minerals were significantly ($P<0.05$) positively correlated with each other, while PA was negatively correlated with all minerals, especially Zn ($P<0.05$) (**Table 7**).

## Discussion

This study data proved the testing hypothesis that current dry pea cultivars and advanced breeding lines bred for conventional systems varied in grain yield and nutritional quality with response to the organic cropping systems. Some of these current dry pea cultivars in production are suitable for the organic production system. "AAC Carver," "Jetset," and "Mystique" are the highest yielding dry pea cultivars (above 2000 kg/ha) and are the most suitable for organic production without a yield penalty (**Fig 1**). The average crude protein content of the cultivars studied is ~21.1 g/100 g, with "CDC Striker" being the highest and "AAC Carver" the lowest (**Fig 1**). Our on-farm organic field trials provide a thorough evaluation of available dry pea cultivars for yield, protein, and other nutritional traits for two years. Organic dry pea grain yields in the present study significantly varied with cultivar, year, and the interaction of cultivar × location ($P<0.05$), indicating cultivar performance is subject to growing conditions, e.g., soil, weather, and organic management conditions. Average dry pea grain yield (769–

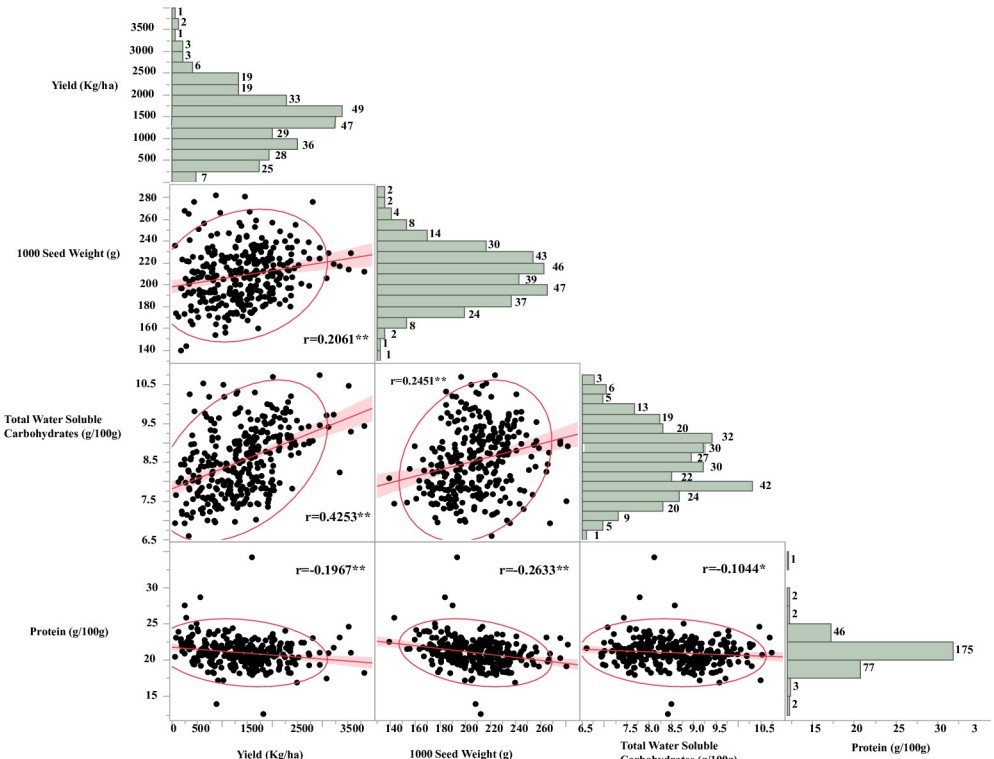

**Fig 3. Correlations and distribution of grain yield, 1000 seed weight, total water-soluble carbohydrates, and protein concentration among the genotypes grown under organic field conditions.**

**Table 6. Correlation of yield, prebiotic carbohydrates, and protein content of organic dry pea genotypes.**

| Variable | Yield | Myo | Xyl | Gal | Sor | Man | Glu | Fru | Suc | Ara | Mal | Sta+Raf | Ver+Kes | Nys | RS | SS | TS | Pro |
|---|---|---|---|---|---|---|---|---|---|---|---|---|---|---|---|---|---|---|
| Yield | - | | | | | | | | | | | | | | | | | |
| Myo-Inositol (Myo) | NS | - | | | | | | | | | | | | | | | | |
| Xylitol (Xyl) | -** | NS | - | | | | | | | | | | | | | | | |
| Galactinol (Gal) | ** | ** | -** | - | | | | | | | | | | | | | | |
| Sorbitol (Sor) | ** | ** | -** | ** | - | | | | | | | | | | | | | |
| Mannitol (Man) | -** | ** | ** | -** | NS | - | | | | | | | | | | | | |
| Glucose (Glu) | ** | ** | NS | ** | ** | NS | - | | | | | | | | | | | |
| Fructose (Fru) | * | NS | NS | NS | ** | ** | ** | - | | | | | | | | | | |
| Sucrose (Suc) | -** | ** | ** | NS | NS | NS | ** | ** | - | | | | | | | | | |
| Arabinose (Ara) | -** | NS | ** | -** | NS | ** | NS | ** | ** | - | | | | | | | | |
| Maltose (Mal) | -** | ** | * | ** | ** | * | ** | ** | ** | ** | - | | | | | | | |
| Sta+Raf | ** | ** | NS | ** | ** | -** | ** | ** | ** | -** | ** | - | | | | | | |
| Ver+Kes | ** | -** | -** | NS | ** | NS | NS | ** | ** | -** | -** | ** | - | | | | | |
| Nystose (Nys) | ** | -** | -** | ** | NS | -** | ** | ** | -** | NS | NS | ** | ** | - | | | | |
| Resistant starch (RS) | -** | -** | ** | ** | ** | ** | -** | -** | ** | NS | ** | ** | -** | ** | - | | | |
| Soluble starch (SS) | ** | -** | NS | ** | ** | ** | ** | ** | NS | NS | NS | -** | ** | -** | -** | - | | |
| Total starch (TS) | ** | ** | ** | -** | ** | NS | ** | NS | NS | NS | NS | ** | NS | NS | NS | ** | - | |
| Protein (Pro) | -** | * | * | NS | NS | ** | NS | NS | * | ** | ** | NS | -** | -** | NS | -* | NS | - |

Stachyose and Raffinose (Sta+Raf); Verbascose and Kestose (Ver+Kes)

** significant at *P<0.05*

* significant at *P<0.1*; Not significant (NS).

2638 kg ha$^{-1}$) and protein concentrations (19.3–24.2 mg/100 g) from this study are similar to results reported for lentil (*Lens culinaris* L.) grown in Canada and dry pea produced in Australia [28, 34, 35]. These current field data are beneficial for future organic dry pea cultivar development for selecting appropriate parents for organic systems to increase grain yield and nutritional quality. Grain yield is an integral part of the organic cultivar development with resistance to abiotic and biotic stresses. As indicated in our data, several of these cultivars have stable yields in organic conditions; still, it is essential to test this hypothesis to ensure that they have had high and stable yields in organic farming conditions over the years [28, 36].

Pulse crops show great potential for biofortification and are suitable for meeting increasing consumer demand for organic plant-based protein, prebiotic carbohydrates, and minerals,

**Table 7. Correlation of yield, critical minerals, and phytic acid concentrations of organic dry pea genotypes.**

| Variable | Yield | K | Mg | Fe | Zn | P | phytic acid |
|---|---|---|---|---|---|---|---|
| Yield | - | | | | | | |
| K | -* | - | | | | | |
| Mg | ** | ** | - | | | | |
| Fe | * | ** | ** | - | | | |
| Zn | NS | ** | ** | ** | - | | |
| P | NS | ** | ** | ** | ** | - | |
| Phytic acid | NS | -* | -* | -* | -** | -* | - |

** significant at *P<0.05*

* significant at *P<0.1*

especially within allergen- and gluten-free markets [7, 37, 38]. Our results indicate organic dry peas are rich in prebiotic carbohydrates (12.5–19.8 g/100 g), providing 63–99% of the RDA for adults (**Table 5**). Sugar alcohols and RFOs have moderate to high broad-sense heritability (0.42–0.75) estimates, indicating it is possible to breed for variable concentrations of these prebiotic carbohydrates for better human health. Sucrose and arabinose are heritable traits, but starch polysaccharides are not (**Table 4**). Total water-soluble carbohydrates (carbohydrates without starch polysaccharides) are significantly and positively correlated with grain yield and 1000-seed weight but negatively correlated with seed protein content (**Fig 3**). This study reported organic dry pea prebiotic carbohydrate concentrations are similar to the values reported in previous data on lentil, dry pea, and chickpea (*Cicer arietinum* L.) [30, 39–42]. Prebiotic carbohydrates are critical components in healthy diets, supporting healthful hindgut microflora. Healthy gut microbiota decreases host obesity, inflammatory bowel diseases, and colorectal cancers and modulates immunological functions by affecting the growth and functioning of host cells [30, 43]. Due to the dietary nature of human metabolic disorders related to obesity, solutions will necessarily focus on a diet–*i.e.*, a cup of pulses a day provides 13–15 g of prebiotic carbohydrates and a range of micronutrients [6, 9, 44]. Changing the levels of these prebiotic carbohydrates is possible by developing molecular markers for marker-assisted breeding with conventional breeding methods in pulse crops; however, genome-wide association mapping studies with diverse populations at several field locations are essential to avoid the yield and protein penalty by changing certain carbohydrates as a result of the quantitative nature of these nutritional traits [37, 45].

Pulses crops, including dry pea, also known as "poor man's meat," are low in fat and provide significant quantities of dietary protein (20–25 g/100 g) and minerals [7, 46]. A 50-g serving of conventional grown dry pea provides 3.7–4.5 mg of Fe, 2.2–2.7 mg of Zn, and 22–34 μg of Se and is very low in PA (2.5–4.4 mg g$^{-1}$), which decreases the bioavailability of minerals [5, 6]. Similar to previous studies, our results show organic dry peas are also rich in Fe, Zn, and Se but not a good Ca source (**Table 5**). Integrating genome-wide research approaches with conventional plant breeding to identify genetic markers associated with these mineral traits could significantly accelerate biofortification efforts by enabling molecular screening of exotic germplasm collections and elite cultivars [23, 37]. No research has been conducted regarding reducing the yield gap without compromising nutritional yield and developing genomic tools for marker-assisted breeding of organic pulse cultivars, i.e., biofortification of organic pulse grains. Therefore, it is essential within the organic farming framework to focus on organic plant breeding activities that will result in cultivars that are more suitable for organic production environments and deliver economic and social benefits to growers and consumers.

Overall, organic markets (especially the gluten-free market) will continue to grow >10–20% per annum at the retail sales level for the foreseeable future in all food categories due to increasing awareness of the connection between diet and human health [12]. Successful organic pulse crop production would increase regional acreage, grower profitability, and stakeholder confidence in organic farming systems. On-farm evaluation of dry pea cultivars and advanced breeding lines under organic management provides valuable information for growers, allowing them to make critical decisions regarding variety selection for (1) growing location, (2) organic management practice, and (3) intended end-use or nutritional quality (prebiotic carbohydrates, protein, minerals, and low phytate), all of which are critical for maximizing grower productivity, profitability, and socio-economic status. Finally, organic dry pea production has potential as a winter cash crop in southern climates; this can be accomplished by selecting diverse genetic material and location sourcing to develop improved cultivars with a higher yield, disease resistance, and nutritional quality.

## Conclusions

Organic dry pea is a potential winter crop in southern US regions. Dry pea grain yields and protein concentrations are within the range of conventional production systems. Organic dry peas are rich in prebiotic carbohydrates (14.7–26.6 g/100 g) and minerals with low concentrations of phytic acid. "AAC Carver," "Jetset," and "Mystique" demonstrated the highest yields and "CDC Striker" the highest protein concentration. These cultivars can be incorporated into organic dry pea breeding programs to develop cultivars suitable for organic production.

## Supporting information

**S1 File. This file is containing all the nutritional phenotyping data used for this manuscript.**
(XLSX)

## Acknowledgments

We thank Bradley Stancil (Crop Improvement, Clemson University), David Robb (Organic Farm, Clemson University), Sorghum Breeding and Genetics (Drs. Boyles and Kresovich team), and Charles Wingard and Ben Dubard from WP Rawl & Sons, Inc., for field operations; and Martin Hochhalter (Meridian Seeds), Tyler Kres (Pulse USA), the Washington State Crop Improvement Association, and the USDA-ARS for providing dry pea seeds.

## Author Contributions

**Conceptualization:** Dil Thavarajah, Rebecca McGee, Shiv Kumar, Rick Boyles.

**Data curation:** Dil Thavarajah, Tristan J. Lawrence, Joshua Kay, Pushparajah Thavarajah, Emerson Shipe.

**Formal analysis:** Dil Thavarajah, Emerson Shipe, Shiv Kumar.

**Funding acquisition:** Dil Thavarajah, Rebecca McGee, Rick Boyles.

**Investigation:** Dil Thavarajah, Rebecca McGee, Rick Boyles.

**Methodology:** Dil Thavarajah, Tristan J. Lawrence, Pushparajah Thavarajah, Shiv Kumar, Rick Boyles.

**Project administration:** Dil Thavarajah, Emerson Shipe, Rick Boyles.

**Resources:** Dil Thavarajah.

**Software:** Dil Thavarajah.

**Supervision:** Dil Thavarajah.

**Validation:** Dil Thavarajah.

**Visualization:** Dil Thavarajah.

**Writing – original draft:** Dil Thavarajah, Sarah E. Powers.

**Writing – review & editing:** Dil Thavarajah, Sarah E. Powers, Pushparajah Thavarajah, Rebecca McGee, Shiv Kumar, Rick Boyles.

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
