## [Decision Letter · Decision Letter 0]

1 Oct 2021

PONE-D-21-24900Organic dry pea (Pisum sativum L.) biofortification for better human healthPLOS ONE

Dear Dr. Thavarajah,

Thank you for submitting your manuscript to PLOS ONE. After careful consideration, we feel that it has merit but does not fully meet PLOS ONE’s publication criteria as it currently stands. Therefore, we invite you to submit a revised version of the manuscript that addresses the points raised during the review process.

I have received evaluations of your manuscript and the reviewers have recommended acceptance for publication after major revision. Please carefully consider the comments and suggestions of the reviewers while revising the manuscript.

We look forward to receiving your revised manuscript.

Kind regards,

Ahmad Naeem Shahzad

Academic Editor

PLOS ONE

“Funding support for this project was provided by the Organic Agriculture Research and Extension Initiative (OREI) (award no. 2018-51300-28431/proposal no. 2018-02799) of the United States Department of Agriculture, National Institute of Food and Agriculture, and the USDA-ARS Pulse Health Initiative (DT and RB); the Specialty Crop Block Grant Program of the U.S. Department of Agriculture through grant AM180100XXXXG026 (DT); the USDA National Institute of Food and Agriculture, [Hatch] project [1022664] (DT); and the Good Food Institute (DT).”

Reviewers' comments:

Reviewer's Responses to Questions

**Comments to the Author**

1. Is the manuscript technically sound, and do the data support the conclusions?

Reviewer #1: Yes

Reviewer #2: Partly

Reviewer #3: Yes

2. Has the statistical analysis been performed appropriately and rigorously? 

Reviewer #1: Yes

Reviewer #2: Yes

Reviewer #3: No

3. Have the authors made all data underlying the findings in their manuscript fully available?

Reviewer #1: Yes

Reviewer #2: Yes

Reviewer #3: Yes

4. Is the manuscript presented in an intelligible fashion and written in standard English?

Reviewer #1: Yes

Reviewer #2: No

Reviewer #3: Yes

5. Review Comments to the Author

Reviewer #1: I have evaluated the manuscript (PONE-D-21-24900) entitled “Organic dry pea (Pisum sativum L.) biofortification for better human health” submitted for publication in ‘PLoS ONE’. The idea of study is interesting and falls within the scope of journal, and can be considered for publication after major revisions as suggested below:

1. In abstract section, add name of few best performing cultivars or breeding lines to develop 30 biofortified organic cultivars with increased yield and nutritional quality.

2. Overall manuscript is written in good English but needs proof reading to remove minor typo mistakes. The authors should be very specific in using abbreviations; write full in once and then use abbreviations. For instance, in whole manuscript N and nitrogen is written consistently. Phosphorus is written as P in line 68 but abbreviated in line 79. This issue with other abbreviations is consistent in whole manuscript, so check it carefully. Moreover, avoid use of uncommon abbreviations like TSW rather than 1000-seed weight.

3. Use SI units in whole manuscript. For instance, the authors use acers units rather than hectare in paragraph starting from line 94. Likewise at line 139 write 0-6 inch rather than 0-15 cm. Check whole manuscript and use SI units. Likewise, kg/ha should be kg ha-1.

4. In paragraph starting from lines 99-103, the authors write references in different style as (19) and (20). Use same format according to journal.

5. Add scientific name of each crop at its 1st occurrence. For example, scientific name of all crops mentioned in lines 86 and 97 are missing. Check whole manuscript.

6. Add testing hypothesis at the end of introduction section.

7. The authors stated that they conducted experiment at two locations during 2019 with two replications and at one location with three replications during 2020. Why different number of locations and replications are used in both years of study? Moreover, two replications are not enough. Also change sub-heading field design to experimental details. Add exact sowing and harvesting dates of both years.

8. Details of weather conditions and soil fertility status prior to sowing should be shifted in Materials and Methods section.

9. Discussion section is written very poorly and full of results repetition and irrelevant information. For instance, no need of introductory paragraph (lines 290-296) in this section. Likewise, paragraph (lines 306-314) regarding weeds control other paragraph (lines 355-367) about crop rotations is not needed here because the authors did not record any weeds data and no crop rotation treatments was included. Be very specific in this section and focus only on results of this study. At start, mention either the hypothesis was accepted or not and then explain results logically without repetition. Try to correlate the results for better understanding and update literature cited.

10. Conclusion section is too long. Add concrete conclusion based on current study results only just within 3-4 sentences.

Reviewer #2: The manuscript introduction lacks the hypothesis why the pea’s varieties for biofortification were assessed, no back ground on data supporting it, for instance previous work if any on biofortification of peas especially under organic farming. Literature on US system for pea cultivation should be reduced and relevant literature on potential area under organic dry cultivation in different parts of world should be added. At moment, reading manuscript is limiting its scope to regional or to specific geographical area.

In materials and methods, no details are provided, how organically crop was managed, as authors reported that no organic fertilizers were used. Details on crop husbandry practices should be detailed for validation and replication while these also be discussed independently from land preparation.

Details on collection of data on seed traits should be presented under subheading.

Details on protocols for measuring seed quality traits should be reduced.

No details on minerals determination and phytic acid contents have been added.

In results, Independently ANOVA results should be discussed first and then their variation over years, locations and interactions.

In discussion part, lines 301-305, rather than discussion and supporting results with possible reasons, authors have discussed the possible implications or future endeavors, how these results may be beneficial that may be part of conclusion

A line 306-314 discusses the weed control strategies, however, this was not hypothesis, and then such un-necessary details should be removed.

Lines 315-320 discuss the improvement in protein contents but does not supported by particular reason for improvement.

In general, discussion need drastic improvement, as it is mainly based on speculations rather than supported with pertinent literature.

Details of Table 1 for example locations, growing years and replicates should be added into experimental design

Only here details of genotypes, breeding lines should be added.

Reviewer #3: Authors have provided the useful information to growers to select or develop cultivars for organic production as these can be a winter cash crop in southern climates. The manuscript can be accepted after major revision.

Please see my comments and/or suggestions in the attached file.

6. PLOS authors have the option to publish the peer review history of their article (what does this mean?). If published, this will include your full peer review and any attached files.

Reviewer #1: No

Reviewer #2: **Yes: **Hafeez ur Rehman

Reviewer #3: No

---

## [Author Response · Author response to Decision Letter 0]

6 Nov 2021

Response to Reviewers

Editors’ comments

1. Comment: Please ensure that your manuscript meets PLOS ONE's style requirements, including those for file naming. The PLOS ONE style templates can be found at https://journals.plos.org/plosone/s/file?id=wjVg/PLOSOne_formatting_sample_main_body.pdf and https://journals.plos.org/plosone/s/file?id=ba62/PLOSOne_formatting_sample_title_authors_affiliations.pdf

 Response: Action has been taken. 

2. Comment: We note that the grant information you provided in the ‘Funding Information’ and ‘Financial Disclosure’ sections do not match. When you resubmit, please ensure that you provide the correct grant numbers for the awards you received for your study in the ‘Funding Information’ section.

Response: Corrected. 

3. Comment: Thank you for stating the following in the Acknowledgments Section of your manuscript: “Funding support for this project was provided by the Organic Agriculture Research and Extension Initiative (OREI) (award no. 2018-51300-28431/proposal no. 2018-02799) of the United States Department of Agriculture, National Institute of Food and Agriculture, and the USDA-ARS Pulse Health Initiative (DT and RB); the Specialty Crop Block Grant Program of the U.S. Department of Agriculture through grant AM180100XXXXG026 (DT); the USDA National Institute of Food and Agriculture, [Hatch] project [1022664] (DT); and the Good Food Institute (DT).”

We note that you have provided additional information within the Acknowledgements Section that is not currently declared in your Funding Statement. Please note that funding information should not appear in the Acknowledgments section or other areas of your manuscript. 

We will only publish funding information present in the Funding Statement section of the online submission form. Please remove any funding-related text from the manuscript and let us know how you would like to update your Funding Statement. Currently, your Funding Statement reads as follows: “The funders had no role in study design, data collection and analysis, decision to publish, or preparation of the manuscript. ”Please include your amended statements within your cover letter; we will change the online submission form on your behalf.

Response – Corrected. I removed the funding sources from the acknowledgments and please see the funding statement below. 

Funding statement: “The funders had no role in study design, data collection and analysis, decision to publish, or preparation of the manuscript. Its contents are solely the responsibility of the authors and do not necessarily represent the official views of the USDA ”

Funding sources: Funding support for this project was provided by the Organic Agriculture Research and Extension Initiative (OREI) (award no. 2018-51300-28431/proposal no. 2018-02799) of the United States Department of Agriculture, National Institute of Food and Agriculture (DT and RB), and the USDA-ARS Pulse Health Initiative (DT); the Specialty Crop Block Grant Program of the U.S. Department of Agriculture through grant AM180100XXXXG026 (DT); the USDA National Institute of Food and Agriculture, [Hatch] project [1022664] (DT); and the Good Food Institute (DT)..

4. Comment: In your Data Availability statement, you have not specified where the minimal data set underlying the results described in your manuscript can be found. PLOS defines a study's minimal data set as the underlying data used to reach the conclusions drawn in the manuscript and any additional data required to replicate the reported study findings in their entirety. All PLOS journals require that the minimal data set be made fully available. For more information about our data policy, please see http://journals.plos.org/plosone/s/data-availability. Upon re-submitting your revised manuscript, please upload your study’s minimal underlying data set as either Supporting Information files or to a stable, public repository and include the relevant URLs, DOIs, or accession numbers within your revised cover letter. For a list of acceptable repositories, please see http://journals.plos.org/plosone/s/data-availability#loc-recommended-repositories. Any potentially identifying patient information must be fully anonymized. Important: If there are ethical or legal restrictions to sharing your data publicly, please explain these restrictions in detail. Please see our guidelines for more information on what we consider unacceptable restrictions to publicly sharing data: http://journals.plos.org/plosone/s/data-availability#loc-unacceptable-data-access-restrictions. Note that it is not acceptable for the authors to be the sole named individuals responsible for ensuring data access. We will update your Data Availability statement to reflect the information you provide in your cover letter.

 Response: This study's minimal underlying data set has been submitted as Supporting Information files.

Reviewers' comments:

Reviewer's Responses to Questions

Comments to the Author

1. Is the manuscript technically sound, and do the data support the conclusions?

Reviewer #1: Yes

Reviewer #2: Partly

Reviewer #3: Yes

2. Has the statistical analysis been performed appropriately and rigorously? 

Reviewer #1: Yes

Reviewer #2: Yes

Reviewer #3: No

3. Have the authors made all data underlying the findings in their manuscript fully available?

Reviewer #1: Yes

Reviewer #2: Yes

Reviewer #3: Yes

4. Is the manuscript presented in an intelligible fashion and written in standard English?

Reviewer #1: Yes

Reviewer #2: No

Reviewer #3: Yes

5. Review Comments to the Author

Reviewer 1: 

Reviewer #1: I have evaluated the manuscript (PONE-D-21-24900) entitled “Organic dry pea (Pisum sativum L.) biofortification for better human health” submitted for publication in ‘PLoS ONE’. The idea of study is interesting and falls within the scope of journal, and can be considered for publication after major revisions as suggested below:

1. Comment: In abstract section, add name of few best performing cultivars or breeding lines to develop 30 biofortified organic cultivars with increased yield and nutritional quality.

Response: Action was taken – see lines 43-46. 

2. Comment: Overall manuscript is written in good English but needs proof reading to remove minor typo mistakes. The authors should be very specific in using abbreviations; write full in once and then use abbreviations. For instance, in whole manuscript N and nitrogen is written consistently. Phosphorus is written as P in line 68 but abbreviated in line 79. This issue with other abbreviations is consistent in whole manuscript, so check it carefully. Moreover, avoid use of uncommon abbreviations like TSW rather than 1000-seed weight.

Response: Action was taken – the manuscript was proofed. 

3. Comments: Use SI units in whole manuscript. For instance, the authors use acers units rather than hectare in paragraph starting from line 94. Likewise at line 139 write 0-6 inch rather than 0-15 cm. Check whole manuscript and use SI units. Likewise, kg/ha should be kg ha-1.

Response: Action was taken – the manuscript was proofed and corrected. 

4. Comment: In paragraph starting from lines 99-103, the authors write references in different style as (19) and (20). Use same format according to journal.

Response: Action was taken – I am so sorry about this matter; it was a mistake from my reference database. I changed all the references for Vancouver style. 

5. Comment: Add scientific name of each crop at its 1st occurrence. For example, scientific name of all crops mentioned in lines 86 and 97 are missing. Check whole manuscript.

Response: Action was taken – the manuscript was corrected. 

6. Comment: Add testing hypothesis at the end of introduction section.

Response: Action was taken – the manuscript was corrected. 

7. Comment: The authors stated that they conducted experiment at two locations during 2019 with two replications and at one location with three replications during 2020. Why different number of locations and replications are used in both years of study? Moreover, two replications are not enough. Also change sub-heading field design to experimental details. Add exact sowing and harvesting dates of both years.

Response: I agree that to increase the experimental power, we need more replications and locations. However, seeds were minimal; therefore, we decided to go with the two replicates and sites for the first year. We had an issue with planting in the second location in the second year due to the severe flood and rain in 2020. We missed the planting date because of the weather advisory. Line 137 has been added. Yes, I changed the sub-headings for the field experiment design and added exact planting dates (lines 158-161). 

8. Comment: Details of weather conditions and soil fertility status prior to sowing should be shifted in Materials and Methods section.

Response: It was removed from the methods and materials section. 

9. Comment: Discussion section is written very poorly and full of results repetition and irrelevant information. For instance, no need of introductory paragraph (lines 290-296) in this section. Likewise, paragraph (lines 306-314) regarding weeds control other paragraph (lines 355-367) about crop rotations is not needed here because the authors did not record any weeds data and no crop rotation treatments was included. Be very specific in this section and focus only on results of this study. At start, mention either the hypothesis was accepted or not and then explain results logically without repetition. Try to correlate the results for better understanding and update literature cited.

Response: Thank you for the suggestion – it is fixed as suggested. 

10. Comment: Conclusion section is too long. Add concrete conclusion based on current study results only just within 3-4 sentences.

Response: Thank you for the suggestion – it is fixed as suggested. 

Reviewer #2: 

1. Comment: The manuscript introduction lacks the hypothesis why the pea’s varieties for biofortification were assessed, no back ground on data supporting it, for instance previous work if any on biofortification of peas especially under organic farming. Literature on US system for pea cultivation should be reduced and relevant literature on potential area under organic dry cultivation in different parts of world should be added. At moment, reading manuscript is limiting its scope to regional or to specific geographical area.

Response: Thank you for your comment. The hypothesis was included in the manuscript. I have done more than a decade of work on pulse biofortification for the conventional system, but organic biofortification research is minimal. I have added a paragraph to the introduction. 

2. Comment: In materials and methods, no details are provided, how organically crop was managed, as authors reported that no organic fertilizers were used. Details on crop husbandry practices should be detailed for validation and replication while these also be discussed independently from land preparation. Details on collection of data on seed traits should be presented under subheading.

Response: The Previous crop was included (lines 153-155). All these on-farn fields are USDA-certified land, and the certification protocol should be available from the USDA. Also, organic management is country-specific; therefore, it is hard to replicate the same certificate process for organic systems. We did not use any management practices except the weed control by machine or manual. No chemicals were used to control any diseases etc. 

3. Comment: Details on protocols for measuring seed quality traits should be reduced.

Response: Details were added. I am sorry for this oversight. 

4. Comment: No details on minerals determination and phytic acid contents have been added.

Response: Analysis methods were included. 

5. Comment: In results, Independently ANOVA results should be discussed first and then their variation over years, locations and interactions.

Response: Thank you for the comment. It is hard to separate by each component as we are discussing several nutritional traits. I add a sentence at the beginning to explain the cultivar effect. 

6. Comment: In discussion part, lines 301-305, rather than discussion and supporting results with possible reasons, authors have discussed the possible implications or future endeavors, how these results may be beneficial that may be part of conclusion.

Response: Thank you for the response. It was removed. 

7. Comment: A line 306-314 discusses the weed control strategies, however, this was not hypothesis, and then such un-necessary details should be removed.

Response: Thank you for the response. It was removed.

8. Comment: Lines 315-320 discuss the improvement in protein contents but does not supported by particular reason for improvement. In general, discussion need drastic improvement, as it is mainly based on speculations rather than supported with pertinent literature.

Response: Discussion has chaged as requested by other reviewers. Thank you for the comment. 

9. Comment: Details of Table 1 for example locations, growing years and replicates should be added into experimental design. Only here details of genotypes, breeding lines should be added.

Response: This comment is not very clear. Table 1 has all the details and is explained under the experimental design – lines 137-140. 

Reviewer #3: Authors have provided the useful information to growers to select or develop cultivars for organic production as these can be a winter cash crop in southern climates. The manuscript can be accepted after major revision.

Please see my comments and/or suggestions in the attached file.

Comment: Author are using mineral data from the National Institute of Health. These data can be used to support their results. Is this appropriate to use in the abstract and results? Same with phytic acid. I did not find this in the methods section. Please indicate source of data.

Response: Sorry for the confusion. We measured minerals using ICP_EMS and phytic acid using HPLC, which is included in the method section. 

Comment: I have highlighted numbers at the various places. Please check and replace them with references.

Response: All the references were fixed. 

Comment: Authors should provide details of modifications.

Response: We listed everything we did in the procedure. It is modified means that we did not buy their commercial kit. It is all detailed in the procedure. 

Comment: Based on the results (Table 4), effect of year and locations had a significant effect for the majority of traits. Data should be combined only if they are not significant. Please justify.

Response: Our first year, we had only two reps due to the seed limitation. We did both statistical analyses by location, year, and combined analysis. After consulting a statistician, we highlighted the cultivars that outperformance regardless of the year and location. We select some candidate parents by combining data when we do not have enough to replicate sites and years in the experiment. Thank you for pointing this out, but we test data for homogeneity. I guess this is another way of presenting this data. 

Comment: Should this be significant? 

For significance, P value should be <0.05.

Please check this in the entire text including Tables.

Response: We set the P-value for 5% and 10%. That is P<0.05 and P<0.1. Since this is an organic field experiment, we decided to go for a higher P-value. 

Comment: Did authors calculate protein based on plots? They can recommend cultivars based on yield as well as protein. 

Response: Great suggestion, but in this study, we measured only protein g/100g. 

6. PLOS authors have the option to publish the peer review history of their article (what does this mean?). If published, this will include your full peer review and any attached files.

Do you want your identity to be public for this peer review? For information about this choice, including consent withdrawal, please see our Privacy Policy.

Reviewer #1: No

Reviewer #2: Yes: Hafeez ur Rehman

Reviewer #3: No

---

## [Editor Report · Decision Letter 1]

25 Nov 2021

Organic dry pea (Pisum sativum L.) biofortification for better human health

PONE-D-21-24900R1

Dear Dr. Thavarajah,

We’re pleased to inform you that your manuscript has been judged scientifically suitable for publication and will be formally accepted for publication once it meets all outstanding technical requirements.

Kind regards,

Ahmad Naeem Shahzad

Academic Editor

PLOS ONE